# Dietary Intake and Sources of Added Sugars in Various Food Environments in Costa Rican Adolescents

**DOI:** 10.3390/nu14050959

**Published:** 2022-02-24

**Authors:** Rafael Monge-Rojas, Rulamán Vargas-Quesada, Uriyoán Colón-Ramos, Anne Chinnock

**Affiliations:** 1Nutrition and Health Unit, Costa Rican Institute for Research and Education on Nutrition and Health (INCIENSA), Ministry of Health, Tres Ríos 4-2250, Costa Rica; 2Department of Human Nutrition, Campus Rodrigo Facio, Universidad de Costa Rica, San José 2060, Costa Rica; rulavargas@gmail.com (R.V.-Q.); anne.chinnock@ucr.ac.cr (A.C.); 3Department of Global Health, Milken Institute School of Public Health, George Washington University, Washington, DC 20052, USA; uriyoan@gwu.edu

**Keywords:** adolescents, added sugar, home environment, school environment, neighborhood environment, Costa Rica

## Abstract

Consumption of added sugars, especially from sugar-sweetened beverages (SSBs), has been associated with several negative health outcomes during adolescence. This study aimed to identify dietary intake and food sources of added sugars in the home, school, and neighborhood environments of Costa Rican adolescents. Dietary intake of added sugars was determined using 3-day food records in a cross-sectional study of 818 adolescents aged 12 to 19 and enrolled in rural and urban schools in the province of San José. On average, 90% of adolescents consumed more than 10% of their total energy intake from added sugars. Furthermore, 74.0% of added sugars were provided at home, 17.4% at school, and 8.6% in the neighborhood. Added sugars were primarily provided by frescos (29.4%), fruit-flavored still drinks (22.9%), and sugar-sweetened carbonated beverages (12.3%), for a total contribution of 64.6%. Our findings suggest that Costa Rican adolescents have a plethora of added sugar sources in all food environments where they socialize. However, it is relevant for public health to consider the home and school environments as fundamental units of interventions aimed at reducing added sugars in the adolescent diet. Frescos prepared at home and school and fruit-flavored still drinks must be the focus of these interventions.

## 1. Introduction

Added sugars are defined as any caloric sweeteners, including honey and syrups, added to food during processing/manufacturing or at the table [1]. Consumption of added sugars, especially from sugar-sweetened beverages (SSBs), has been associated with increased risk of weight gain [2,3,4,5], insulin resistance [6,7], metabolic syndrome [8,9,10], lipid panel abnormalities [1,2,7,8,9,11], trunk fat accumulation, obesity [10,12] and visceral adiposity [8]. Adolescents living in low- and middle-income countries are a particularly vulnerable group to high consumption of added sugars [13]. Several studies indicate that approximately 80% of adolescents consume well over 10% of their total daily calories from added sugars [14,15,16,17,18,19,20,21].

Public health actions targeted at reducing consumption of added sugars among adolescents, especially from SSBs, have increased over the past two decades. Some [22,23,24] but not all studies [25,26,27,28] point to the school environment as an efficient platform to promote this reduction, especially given the positive association between student purchases from school outlets and SSB intake [22]. Nonetheless, other studies suggest that the home environment is a more significant contributor to SSB consumption. It has been documented that adolescents with more frequent availability of SSBs at home have a higher consumption of these drinks [29,30,31,32], regardless of SSB availability in other environments (e.g., school and neighborhood). On the other hand, the universal availability of SSBs in the neighborhood environment contributes to impulse purchasing and consumption among adolescents [33]. Since most adolescents in Costa Rica cannot drive independently, they may be restricted to purchasing or consuming SSBs in areas near their home and school to which they can walk or bike. Hearts et al. [34] found that youth purchases of SSBs in community settings are more frequent when food retail stores are available within a 10 min walk from home. Likewise, Laska et al. [33] found that intake of SSBs in adolescents was associated with residential proximity to restaurants (including fast food), convenience stores, grocery stores, and other retail facilities within the 800 m and 1600 m residential buffer.

The adolescence period has been identified as an important window of opportunity for public health efforts to achieve dietary modifications that enhance health-conscious dietary habits [35,36,37]. Children between the ages of 2 and 9 consume more foods high in added sugars on weekends compared with weekdays [38]. Nonetheless, the patterns of consumption of added sugars and the environments in which this consumption occurs among children over the age of 10, especially in low- and middle-income countries, is not well-documented. This lack of evidence limits the ability of public health efforts to identify and target intervention points in the adolescent food environments. Given the detrimental health consequences [39] and dietary quality impairment [40,41] of a high added sugar intake, it is indispensable to identify added sugar sources in various adolescent food environments and pinpoint critical areas to develop and appropriately target interventions that reduce SSB intake. This study aimed to (1) identify dietary intake and food sources of added sugars in the home, school, and neighborhood environments of Costa Rican adolescents, and (2) determine if added sugar consumption varied according to mealtimes and between weekdays and weekends in this population.

We hypothesized that added sugars in the adolescent diet are primarily consumed at home and come from carbonated SSBs.

## 2. Methods

### 2.1. Study Population and Setting

This study used cross-sectional data from adolescents enrolled in rural and urban schools in the province of San José, Costa Rica, in 2017. San José has the highest adolescent concentration (30%) in the country [42]. In addition, school enrollment is high (80%) in Costa Rica [43].

Eighteen high schools were selected from a list of all the public and private high schools in the San José province. Seventh to eleventh graders (13–18 years old) enrolled in the selected schools were invited to participate in the study. Sample size was determined assuming a sampling error for a population proportion using a 95% confidence interval with a permissible error of 5% and correction for a finite population [44]. Sample selection was carried out in three stages. In the first stage, schools were selected using a proportional-size probability method [45]. In the second stage, 10 classrooms from each school (two from each grade) were selected using simple random sampling. All students were invited to participate in the study. The informed assent form was explained to all interested adolescents. Those who agreed to participate were given informed consent forms to take to their parents/tutors for permission to participate in the research study. In the third stage, participants were chosen randomly from the students who returned signed informed consent and assent forms. Approximately 5% of the initial sample decided not to participate in the study before it started. The final study sample comprised 818 adolescents aged from 12 to 19 years.

### 2.2. Dietary Intake Assessment

Dietary intake data were collected via 3-day food records completed by the participants in real time and reviewed by nutritionists. Students were asked to complete 3-day food records on two weekdays (Monday, Tuesday, Thursday, or Friday) and one weekend day (Saturday or Sunday). Half of the participants were randomly selected to record the foods and drinks they consumed on Thursday, Friday, and Saturday; while the rest were asked to complete the record on Sunday, Monday, and Tuesday. Data were collected during nine months of the school year (February to November), reflecting seasonal variations for Costa Rica: rainy season (May to November) and dry season (December to April). The goal was to ensure that the data captured daily and seasonal variability in food consumption.

At each school, six trained nutritionists provided participants with a notebook divided into three sections (one per day). Each section was subdivided by mealtime (prebreakfast, breakfast, morning snack, lunch, afternoon snack, dinner, bedtime snack). Participants were taught how to complete accurate food records by writing detailed descriptions of what they ate and drank from the time they woke up in the morning to the time they went to bed at night for three consecutive days. Adolescents were asked to write down brand names of foods when appropriate, methods of preparation, and recipes for all dishes and drinks whenever possible. They also had to indicate where the meal took place and the provenance of the food (home, school, or neighborhood). The nutritionists taught the participants to estimate portion sizes using an established serving-size manual developed and validated for Costa Rica [46]. The manual includes full-color photographs and diagrams of typical local foods and their preparation, with 3 to 6 serving sizes per food. Each adolescent received a notebook and manual to estimate and record food intake for three days. Adolescents were instructed to report portion sizes using measurements based on household utensils or volume and mass units.

Given the challenges related to incompleteness and inaccuracy when recording self-reported dietary data in young populations and specific demographic groups [47], the nutritionists thoroughly reviewed the completed 3-day food records, conducting one-on-one interviews with each participant during school hours. At these interviews, the nutritionists inquired about commonly missed items or ingredients (e.g., added sugars, candies, beverages); entered additional details about the types of food or drinks that were consumed; verified or added portion sizes; made sure meal location and food provenance (home, school, or neighborhood) were entered; and clarified any illegible items. The nutritionists used food models, fresh foods, and various utensils to verify portion sizes.

### 2.3. Usual Intake

The 3-day food records were used to estimate usual food consumption and assess intra-individual variability in nutrient intakes. Multiple Source Method (MSM; https://msm.dife.de/tps/en, accessed 14 December 2021), a web-based statistical modeling technique proposed by the European Prospective Investigation into Cancer and Nutrition (EPIC), was used to estimate energy and macronutrient intakes. This method was chosen because of its ability to improve estimates of usual dietary intake of energy and nutrients by considering within-person variance in intake, thereby improving the usual intake distribution for the population [48].

### 2.4. Added Sugar Assessment

Dietary intake of added sugars was estimated using the following resources: (1) recipes provided by the adolescents on their food records, as well as from a collection of recipes from dietary studies conducted in communities, available at the School of Human Nutrition of the University of Costa Rica; (2) the database of the School of Human Nutrition of the University of Costa Rica, which contains 1655 items from the following sources: 1307 foods from USDA, 80 foods from the food composition tables of the Institute of Nutrition for Central America and Panama (INCAP), 254 foods from recipes commonly consumed in Costa Rica (e.g., frescos, pastries, breads, cookies and desserts), 13 foods from the Costa Rican mandatory fortification food group, and updated information on foodstuffs from the food and beverage industry in Costa Rica as of 2017 (year in which the study data were collected), including all the foods that were reformulated by the Costa Rican food industry to reduce added sugar content; (3) manufacturer’s information, and (4) lists of ingredients and other labeling information. Although all these tools were used to estimate consumption of added sugars, it was impossible to collect precise information for every single case; therefore, estimates were made for some food items using values from similar foods, as has been suggested by Wanselius et al. [49].

Added sugar contributors were defined as: all refined or industrially produced sugar (e.g., brown sugar, corn sweetener, corn syrup, dextrose, fructose, glucose, sucrose, high-fructose corn syrup, honey, invert sugar, maltose, malt syrup, molasses, raw sugar, and naturally occurring sugars that are isolated from a whole food and concentrated so that sugar is the primary component) used as an ingredient in processed or prepared foods such as sweets, cookies, dairy products (e.g., ice cream, yogurt, sweetened condensed milk), carbonated drinks, fruit-flavored still drinks (e.g., flavored iced teas, industrialized tropical fruit-flavored drinks, industrialized fruit juices and nectars, isotonic drinks), bakery products, desserts, and breakfast cereals. In addition, all sugar added to frescos (traditional homemade beverages made with fresh fruit juice, sugar, and water) or sugars added at the table (e.g., to coffee) were considered contributors to added sugars. Naturally occurring sugars from fruits (intact or juiced), vegetables (intact or juiced), and milk were not considered to be added sugar contributors.

### 2.5. Comparison with Dietary Recommendations for Added Sugar

Given that the World Health Organization (WHO) does not provide a specific recommendation for added sugars, the analyses used the recommendation of consuming < 10% of total energy intake (TEI) from added sugars established by the 2015–2020 edition of the Dietary Guidelines for Americans (DGA) as a basis for comparison [50]. The WHO recommends limiting consumption of free sugars (all monosaccharides and disaccharides added to foods by the manufacturer, cook, or consumer, plus the sugars that are naturally present in honey, syrups, and fruit juices) to less than 10% of TEI, and ideally to less than 5% of TEI to achieve additional health benefits [51].

### 2.6. Food Sources

A weighted-proportion formula developed by Block et al. [52] was used to determine the contributions of different food groups to added sugar intake.

### 2.7. Data Analysis

Epi Info™ software, version 3.5.4 (2008), was used to process data from the 3-day food records. The previously described database of the School of Human Nutrition of the University of Costa Rica was used with this tool.

The location where adolescents obtained their food was considered as the intake location, since this is the environment providing added sugars to the adolescent diet.

Variance analysis was performed using GLM adjusted by energy intake to determine if added sugar intakes vary across gender and concentration of urbanization. The significance of the differences between male and female means and urban and rural means was assessed using the Tukey multiple comparison test. Differences in the daily means of energy intake, added sugars, and percentage of total energy (TE) from added sugars by sociodemographic characteristics were determined using Student’s *t*-test or the chi-squared test.

Differences between the proportions of added sugars consumed in different social environments, mealtimes, and days of the week were tested using the chi-squared test. The chi-squared test was also used to assess differences between the added sugar contribution of various food groups by sex, residence area, social environment (home, school, neighborhood), and day of the week (weekdays/weekends). Lastly, the chi-squared test was used to assess differences in the percentage of adolescents meeting the added sugar levels established in the 2015 to 2020 DGA [50]. All tests were two-tailed, and the statistical significance level was set at 0.05. Statistical analysis was performed using the Statistical Package for Social Sciences (SPSS Inc., version 27.0 for Windows, Chicago, IL, USA).

## 3. Results

The mean age of the study sample was 15.3 years (1.8), with 63.9% girls, 50.2% urban, and 28.2% overweight/obese adolescents (Table 1). Mean added sugar intake was 93.1 g/d, roughly 19.1% of TEI. Daily intake of foods high in added sugars ranged between 316.6 g/d (frescos) and 9.4 g/d (sweets).

### 3.1. Dietary Intake of Added Sugars

Average TEI from added sugars was higher in urban vs. rural adolescents (19.7% and 18.3% TEI, respectively, *p* < 0.05), as well as in upper and middle vs. lower SES (19.6%, 19.5%, and 17.9% TEI, respectively, *p* < 0.05) (Table 2). Adolescents with overweight/obesity had a higher average TEI from added sugars than those with healthy weights. On weekends, TEI from added sugars was higher than on weekdays (19.8% TEI and 18.9%, respectively, *p* > 0.05). No difference was found in TEI from added sugars between boys and girls (19.0% and 19.3% TEI, respectively, *p* > 0.05).

The percentage of Costa Rican adolescents meeting the dietary recommendations for added sugar intake established by the DGA is presented on Table 3. On average, only 9.9% consumed less than 10% TEI from added sugars. Significantly more rural vs. urban adolescents consumed between 10.1% and 15% TEI from added sugars (24.3% and 17.5% TEI, respectively, *p* < 0.05). In contrast, more urban vs. rural adolescents consumed over 20% TEI from added sugars (47.7% and 38.6% TEI, respectively, *p* < 0.05). A higher proportion of adolescents classified as high SES consumed more than 20% TEI from added sugar than those classified as low SES (46.5% and 36.5% TEI, respectively, *p* < 0.05).

On weekdays, added sugar intake was higher for urban adolescents (*p* < 0.001), and on weekends, it was higher for boys (*p* < 0.05).

### 3.2. Proportion of Added Sugars Consumed in Different Social Environments, Mealtimes, and Days of the Week

Dietary intake of added sugars by social environment, mealtime and day of the week is shown on Figure 1. In total, 74.0% of added sugars consumed by adolescents is provided in the home environment, 17.4% in the school environment, and 8.6% in the neighborhood environment (*p* < 0.05). The meal contributing the most added sugars to the diet was lunch, followed by breakfast and dinner (27.3%, 20.9%, and 18.9%, respectively, *p* < 0.05). Afternoon snacks account for more added sugars than morning and bedtime snacks (18.7%, 10.5%, and 3.8%, respectively, *p* < 0.001). However, the combined added sugar contribution of the three snack mealtimes was significantly higher than lunch (34.3% and 27.3%, respectively, *p* < 0.001). In addition, added sugar intake was significantly lower during weekdays (Monday to Friday) than on weekend days (Saturday and Sunday), at 90.2 g (47.8%) and 100.9 g (69.2%), respectively, *p* < 0.01.

### 3.3. Dietary Intake of Added Sugars by Food Environment and Mealtime by Residence Area

The intake of added sugars provided at home is higher in urban than rural adolescents (*p* < 0.001). The school and neighborhood environments contribute more added sugars to the rural adolescent diet (*p* < 0.01) (Table 4). Added sugar intake from morning snacks and dinner was higher in urban adolescents. The average contribution of added sugars from all snacks was higher for urban residents (*p* < 0.05) and for girls (*p* < 0.001).

### 3.4. Sources of Added Sugars

Added sugars were primarily provided by frescos (29.4%), fruit-flavored still drinks (22.9%), and sugar-sweetened carbonated beverages (12.3%), for a total added sugar contribution of 64.6% (Figure 2). These three foods provided more added sugars in girls (*p* < 0.01). Frescos contributed more added sugars in rural vs. urban adolescents (33.5% and 25.2%, respectively, *p* < 0.0001). Fruit-flavored still drinks and sugar-sweetened carbonated beverages contributed more added sugars in urban adolescents (*p* < 0.01) (Figure 2).

Added sugars provided by cookies and sweets were higher in rural adolescents (*p* < 0.01). In contrast, added sugars provided by desserts, breakfast cereals, fast food, and dairy products were higher in urban adolescents (*p* < 0.05).

Figure 3 presents the added sugar sources in the Costa Rican adolescent diet, categorized by food environment and day of the week. SSBs consumed at school and home provide a similar percentage of added sugars (63.2% and 61.7%, respectively, *p* > 0.05), which is significantly higher than the contribution of the neighborhood environment (54.9%, *p* < 0.05).

At home and school, frescos were the primary source of added sugars (29.6% and 33.2%, respectively, *p* < 0.001). It is worth noting that these sugary frescos are prepared and served at school cafeterias to accompany the students’ lunches. The added sugar contribution of fruit-flavored still drinks was higher in the school (32.2% and 27.1%, respectively *p* < 0.001) vs. the home environment (29.6% and 20.7%, respectively, *p* < 0.001) (Figure 3). At school, fruit-flavored still drinks are purchased at food kiosks. Sugar-sweetened carbonated beverages only contribute < 4% added sugars in the school environment, which is worthy of note.

Fruit-flavored still drinks, sugar-sweetened carbonated beverages, breakfast cereals, bread, and vegetable-milk beverages consumed at home jointly contributed 42.6% of total added sugars. Cookies contributed more added sugars at school than in the home and neighborhood environments.

Sugar-sweetened carbonated beverages, sweets, desserts, bakery products, milk-based beverages, fast food, and dairy products consumed by adolescents in the neighborhood environment jointly contributed more added sugars than other socialization spaces. The contribution of added sugars from sugar-sweetened carbonated beverages, sweets, bakery products, and fast foods was higher on weekends (*p* < 0.01). In contrast, added sugars provided by breakfast cereals, cookies, and fruit-flavored still drinks were higher on weekdays (*p* < 0.01).

## 4. Discussion

This study provides the first picture of added sugar intakes and sources in Costa Rican adolescents. The consumption of added sugar exceeds the recommendation established by the WHO in a large proportion (90%) of adolescents. The highest consumption occurs in the family environment, during breakfast, lunch, and snack mealtimes, and during weekend days.

Like other studies, added sugar intake among Costa Rican adolescents is notably higher than the recommended < 10% TEI [15,18,19,20,21], both on weekdays and weekends. It is of particular concern that, overall, 43% of adolescents consume over 20% TEI from added sugars and that much of this sugar is consumed primarily during breakfast, lunch, and snack mealtimes, and during weekend days. This information is vital for decision makers to identify potential intervention points to define public health strategies that reduce the consumption of added sugars during adolescence.

In accordance with the study’s hypothesis, the home environment was identified as the setting where this sample of Costa Rican adolescents consumed the highest proportion of added sugars, highlighting the critical role parents/tutors continue to play through the foods they make available at home. Numerous studies demonstrate that the home environment can influence healthful eating by restricting the availability of foods high in added sugar [2,3,4,5,31,52,53,54]. These findings further highlight the need for intervention points in the home environment.

Further, like other studies, SSBs were the primary source of added sugars in the Costa Rican adolescent diet, and within those, for this sample, frescos prepared at school and home contributed more added sugars than prepackaged carbonated beverages and fruit-flavored still drinks, which contradicts the study’s hypothesis.

SSBs have been identified as the leading source of added sugars in the adolescent diet [14,15,18,21]. Further, they are associated with cardiometabolic risk factors [1,7,8,9,10,12] and are more strongly associated with obesity and related outcomes than solid sources of added sugars, as they contribute to excess energy intake and weight gain due to low satiety and incomplete compensation at subsequent meals [12,55,56,57]. Various regulatory initiatives to reduce SSB consumption, such as taxes, advertising and marketing restrictions, and product labeling rules, have been proposed and/or adopted in Latin America [58]. Still, these initiatives work under the assumption that industrialized prepackaged SSBs are the primary source of added sugars. Our findings suggest the need to broaden the scope beyond prepackaged SSBs so that water intake is promoted as a substitute of frescos, both at home and at school.

Changing the availability of frescos to promote a healthier diet in adolescents is a complex task because the ingredients to prepare them vary, it is difficult to regulate them in various environments, and their consumption at mealtimes is firmly embedded in the Costa Rican food culture, given the practice of drinking frescos with snacks, lunch and dinner or as thirst-quenchers [59]. It is likely that parents, adolescents, and stakeholders for beverage availability in schools do not attribute any negative connotations to drinking frescos because public health advocates have focused on reducing the consumption of industrialized SSBs, particularly carbonated beverages, thereby reinforcing a negative perception of their harmful health effects [60]. Nonetheless, since frescos are generally fresh-fruit-based, it is likely that parents, adolescents, and policy stakeholders may often fall victim to the ‘health halo effect’ surrounding fruit-based beverages [61,62]; that is, they perceive that frescos are associated with the health benefits of fresh fruits while ignoring their added sugar content. The same thing may happen with fruit-flavored still drinks, which are highly consumed at home and school and whose intake has been less discouraged than carbonated SSBs. Other studies have documented that the concept of ‘natural’ and ‘homemade’ was associated with ‘healthy’ among Latino families in the US [63].

It is important to note that prepackaged carbonated SSBs had a minor contribution to added sugars in the school environment, perhaps reflecting the strategies developed by the Costa Rican Government to restrict SSB availability in this setting [64]. Even if there have been some deficiencies in implementing said strategies, there is evidence of a positive effect of limiting prepackaged carbonated SSBs, despite the availability of other SSBs [65]. Reinforcing the implementation of policies that regulate food availability in the school environment is required to significantly impact the availability and consumption of the various SSBs, as evidenced by several school-based interventions [24].

Our findings suggest that Costa Rican adolescents have a plethora of added sugar sources in all food environments where they socialize. Therefore, public health interventions aimed at reducing added sugar consumption in adolescents must be approached from multiple levels simultaneously, including cultural aspects and geographic and social contexts. The home and school environments must remain the focused targets of such interventions. In developed countries, there is evidence of the effectiveness of school based-interventions [23,66,67,68,69], but studies on the effectiveness of home-based interventions on adolescent SSB intake modification are scarce [70]. Nonetheless, it is relevant for public health to consider these settings as fundamental units of intervention because SSB availability at home is associated with increased adolescent intake [29,30,31,32]. Some home-based interventions that replace SSBs with noncaloric drinks have demonstrated a decreased SSB intake [71,72].

In addition, added sugar content must be reduced in key foods (such as breakfast cereals, breads, cookies, desserts, bakery products, and sweets) that adolescents regularly consume in various food environments. However, the use of artificial sweeteners must be minimized since their consumption has been related to deleterious health effects [73].

Our findings must be interpreted with caution since the sample was not nationally representative; it was limited to urban and rural areas within the province of San José. However, the largest proportion of Costa Rican adolescents (30%) is concentrated in that province [42]. Moreover, the sample was limited to adolescents enrolled in school, though in Costa Rica, around 80% of adolescents are enrolled in high school [43]. Furthermore, the methodology used to estimate added sugar intake may have limitations given the sources of information; however, the actual added sugar intake may be worse than reported. Finally, a strength of this study was the method used for dietary assessment. Although all methods of assessing food consumption have some kind of inherent error, food records are considered the most accurate among the prospective dietary assessment methods. Therefore, despite existing limitations, they have been referred to as an imperfect gold standard [74].

## 5. Conclusions

This study shows that a large proportion of Costa Rican adolescents (90%) considerably surpasses the recommendation to consume less than 10% TEI from added sugars. The home environment is noticeably the main contributor to added sugar intake in adolescents and must be considered a fundamental unit of intervention. Considering that industrialized carbonated SSBs are not the first source of added sugars in the Costa Rican adolescent diet, and based on our findings, the consumption of frescos prepared in the home and school environments and fruit-flavored still drinks must be the focus of any public health strategies aimed at reducing added sugars in the adolescent diet.

## Figures and Tables

**Figure 1 nutrients-14-00959-f001:**
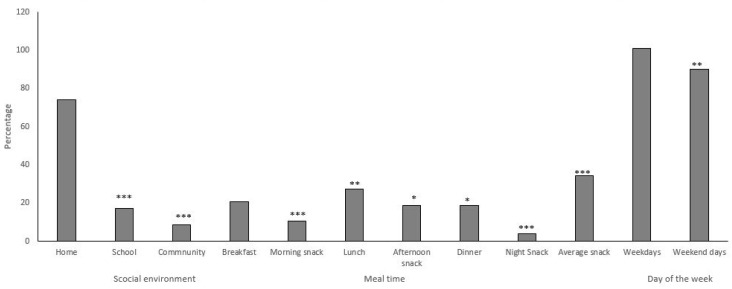
Proportion of added sugars consumed by Costa Rican adolescents in different social environments and mealtimes. Values shown as %. Significant differences determined from chi-squared test shown as * *p* < 0.05, ** *p* < 0.01, *** *p* < 0.0001, as compared to the reference group (home environment, breakfast and weekdays).

**Figure 2 nutrients-14-00959-f002:**
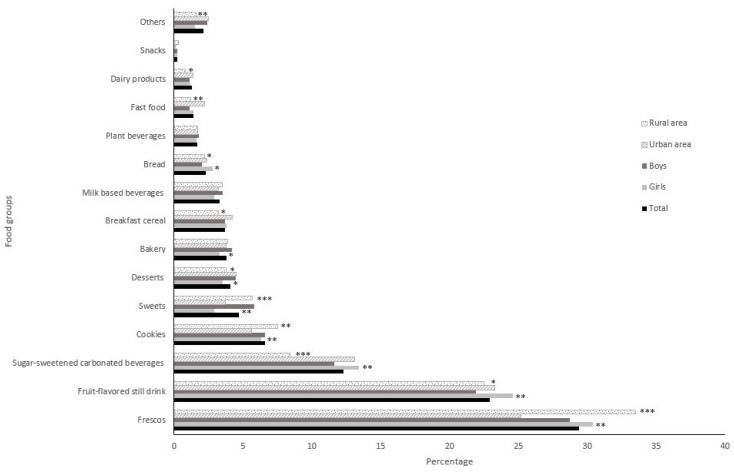
Food sources of added sugars in the Costa Rican adolescent diet, stratified by sex and area. Values shown as %. Significant differences determined from chi-squared test shown as * *p* < 0.05, ** *p* < 0.01, *** *p* < 0.0001, as compared to the reference group (boys and urban area). Fresco is a traditional Costa Rican home-made beverage made by blending pieces of fresh fruit or fresh fruit juice, sugar, and water. Fruit-flavored still drink includes fruit-flavored still waters, iced teas and hydration drinks. Sugar-sweetened carbonated beverages includes Coke, Pepsi, Sprite, and other sodas. Cookies includes all home-made and prepackaged cookies. Sweets includes all sugar confectionery items. Milk based beverages Includes industrialized ready-to-drink flavored milk drinks. Plant beverages includes all type of plant-based milk beverages (cereals, legumes, nuts, seeds, and pseudo cereals). Fast foods includes traditional Costa Rican fast foods (e.g., *empanadas*, *arreglados*, *tacos ticos*) and foods sold in international restaurant chains such as McDonald’s, Burger King, KFC and others. Snacks includes sweet extruded and puffed corn snacks.

**Figure 3 nutrients-14-00959-f003:**
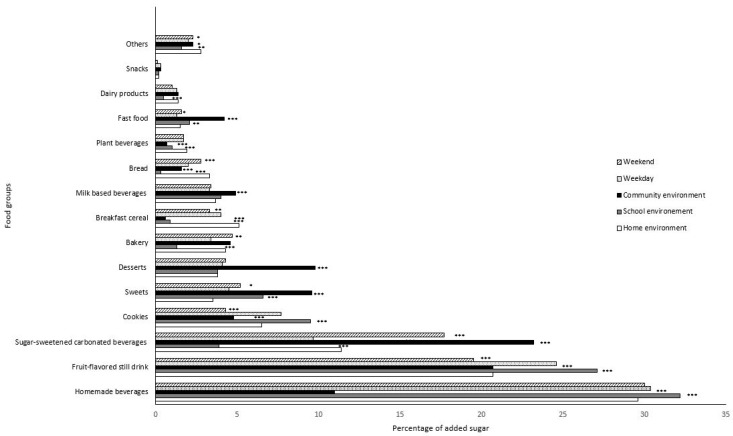
Food sources of added sugars in the Costa Rican adolescents’ diet, stratified by social environment and day of the week. Values shown as %. Significant differences determined from chi-squared test shown as * *p* < 0.05, ** *p* < 0.01, *** *p* < 0.0001, as compared to the referent group (home environment and weekdays). Fresco is a traditional Costa Rican home-made beverage made by blending pieces of fresh fruit or fresh fruit juice, sugar, and water. Fruit-flavored still drink includes fruit-flavored still waters, iced teas and hydration drinks. Sugar-sweetened carbonated beverages includes Coke, Pepsi, Sprite, and other sodas. Cookies includes all home-made and prepackaged cookies. Sweets includes all sugar confectionery items. Milk based beverages includes industrialized ready-to-drink flavored milk drinks. Plant beverages includes all type of plant-based milk beverages (cereals, legumes, nuts, seeds, and pseudo cereals). Fast foods includes traditional Costa Rican fast foods (e.g., empanadas, arreglados, tacos ticos) and foods sold in international restaurant chains such as McDonald’s, Burger King, KFC and others. Snacks includes sweet extruded and puffed corn snacks.

**Table 1 nutrients-14-00959-t001:** General and dietary intake characteristics of the study sample *.

Characteristics	*n* = 818
Study population	
Age (years)	15.3 ± 1.8
Girls (%)	63.9
Urban (%)	50.2
Overweight/Obesity (%)	28.2
Sugar intake (g/d) ^1^	
Total sugar intake (g/d)	114.6 ± 50.5
Added sugar intake (g/d)	93.1 ± 46.3
Intake (g/d) of foods rich in added sugar ^ǂ^	
Sugar-sweetened carbonated beverages (g/d) ^1^	124.5 ± 78.6
Homemade beverages (g/d) ^2^	316.6 ± 171.4
Fruit-flavored still drinks (g/d) ^3^	246.0 ± 110.6
Milk-based beverages (g/d) ^4^	50.7 ± 23.2
Cookies (g/d) ^5^	18.6 ± 8.1
Bakery (g/d)	19.31 ± 9.3
Desserts (g/d)	22.6 ± 10.2
Sweets (g/d) ^6^	9.41 ± 4.5
Snacks and fast foods (g/d) ^7^	38.97 ± 15.7
Bread (g/d)	17.95 ± 8.2
Breakfast cereal (g/d)	55.1 ± 21.4

* Values are means ± SDs or percentages unless otherwise indicated. **^ǂ^** Mean value ± SDs of the consumption (per person per day) of various foods with a high added sugar content, g/d = grams per day. ^1^ Includes Coke, Pepsi, Sprite and other sodas. ^2^ Includes *Fresco* (a traditional Costa Rican home-made beverage made by blending pieces of fresh fruit or fresh fruit juice, sugar, and water) and ready-to-drink coffee. ^3^ Includes fruit-flavored still waters, iced teas and sport drinks. ^4^ Includes prepackaged flavored milk. ^5^ Includes all cookies (home-made and prepackaged). ^6^ Includes all sweet confectionery items. ^7^ Includes sweet extruded and puffed corn snacks, traditional Costa Rican fast foods (e.g., *empanadas*, *arreglados*, *tacos ticos*) and foods sold in international restaurant chains such as McDonald’s, Burger King, KFC and others.

**Table 2 nutrients-14-00959-t002:** Daily means of energy intake, added sugars and percentage of daily TEI from added sugars by sociodemographic characteristics and day of the week.

Variable	Mean Energy Intake (Kcal/d) ^1^	Added Sugar Intake (g/d) ^1^	Percentage of TEI from Added Sugars
Average	1940.2 ± 602.8	93.1 ± 46.3	19.1 ± 7.2
Sex			
Boys	2143.2 ± 602.9	102.1 ± 48.2	19.0 ± 7.1
Girls	1823.5 ± 571.3 ***	87.8 ± 44.4 ***	19.3 ± 7.2
Area			
Urban	1905.6 ± 600.5	95.4 ± 48.1	19.7± 7.2
Rural	1975.3 ± 597.3	90.5 ± 44.3	18.3 ± 7.1
Socioeconomic Level			
Low	1907.5 ± 600.2	84.9 ± 16.6	18.1 ± 6.9
Medium	1972.3 ± 584.3 *	93.9 ± 44.5 **	19.5 ± 7.0 **
High	2016.4 ± 626.9 ***	100.7 ± 52.9 ***	19.6 ± 7.0 ***
Nutritional Status			
Healthy weight	1968.3 ± 605.2	93.2 ± 44.4	18.9 ± 7.1
Overweight	1904.8 ± 507.3	94.1 ± 49.8	19.1 ± 7.5
Obesity	1987.9 ± 591.2	99.2 ± 50.8 *	19.8 ± 7.0 **
Age			
13–15	1913.1 ± 587.3	92.3 ± 44.7	19.1 ± 7.0
16–19	1978.4 ± 622.8	94.2 ± 48.6	18.9 ± 7.3
Days of the week			
Weekdays (Monday-Friday)	1927.2 ± 645.9	90.7 ± 47.8	18.9 ± 6.8
Weekend (Saturday and Sunday)	2095.3 ± 835.6 ***	104.6 ± 43.2 ***	19.8 ± 7.3 **

^1^ Values are means ± SDs. Significant differences determined from *t*-test or chi-squared test shown as * *p* < 0.05, ** *p* < 0.01, *** *p* < 0.0001, compared to the referent group (boys, urban area, low SES, and weekdays). Kcal/d: kilocalories per day; g/d: grams per day; TE: total energy.

**Table 3 nutrients-14-00959-t003:** Percentage of Costa Rican adolescents meeting the U.S. Department of Health and Human Services and U.S. Department of Agriculture dietary recommendations for added sugars.

Percent of Daily Total Energy Intake (TEI) from Added Sugars	Overall (*n* = 818)	Sex		Age Group		Area of Residence		SES		
		Boys (*n* = 298)	Girls (*n* = 520)	12–15 y (*n* = 479)	16–19 y (*n* = 339)	Urban (*n* = 411)	Rural (*n* = 407)	Low (*n* = 263)	Medium (*n* = 325)	High (*n* = 230)
<10% TEI	9.9 (0.3)	7.7 (0.3)	11.2 (0.3)	9.6 (0.3)	10.3 (0.3)	9 (0.3)	10.8 (0.3)	9.9 (0.3)	8.6 (0.3)	11.7 (0.3)
10.1–15% TEI	20.9 (0.4)	24.5 (0.4)	18.8 (0.4)	18.8 (0.4)	23.9 (0.4)	17.5 (0.4)	24.3 (0.4) *	25.9 (0.4)	18.2 (0.4)	19.1 (0.4)
15.1–20% TEI	26 (0.4)	26.5 (0.4)	25.8 (0.4)	28.4 (0.5)	22.7 (0.4)	25.8 (0.4)	26.3 (0.4)	27.8 (0.4)	27.1 (0.4)	22.6 (0.4)
>20% TEI	43.2 (0.5)	41.3 (0.5)	44.2 (0.5)	43.2 (0.5)	43.1 (0.5)	47.7 (0.5)	38.6 (0.5) **	36.5 (0.5)	46.2 (0.5)	46.5 (0.5) *

Values shown as % (SD). Significant differences determined from chi-squared test shown as * *p* < 0.05, ** *p* < 0.01. TEI: Total Energy Intake, SES: Socioeconomic Status.

**Table 4 nutrients-14-00959-t004:** Dietary intake of added sugars (grams/day) in different social environments and mealtimes by area of residence ^1^.

Characteristics	Urban Area		Rural Area		Main Effect *p* Value	
	Boys (*n* = 152)	Girls (*n* = 259)	Boys (*n* = 146)	Girls (*n* = 261)	Gender	Area
Social environment						
Home	80.3 ± 24.8	68.4 ± 21.8	75.1 ± 25.6	61.9 ± 19.7	0.861	<0.001
School	10.4 ± 6.3	10.4 ± 7.2	18.9 ± 9.3	18.5 ± 9.8	0.226	<0.001
Neighborhood	14.2 ± 8.5	11.1 ± 6.4	4.9 ± 2.7	5.3 ± 2.9	0.740	<0.001
Mealtime						
Breakfast	17.9 ± 8.9	15.1 ± 9.7	19.1 ± 10.3	15.7 ± 9.8	0.260	0.672
Morning snack	11.5 ± 7.5	10.8 ± 6.3	8.9 ± 5.4	8.2 ± 6.1	0.133	<0.001
Lunch	27.9 ± 11.2	23.7 ± 10.7	26.7 ± 9.4	21.4 ± 9.9	0.071	0.057
Afternoon snack	16.5 ± 9.2	18.1 ± 8.9	13.7 ± 6.8	19.7 ± 8.7	0.002	0.578
Dinner	22.3 ± 10.1	18.6 ± 9.4	21.2 ± 10.2	15.0 ± 7.2	0.343	0.009
Bedtime snack	4.6 ± 2.3	2.9 ± 1.3	4.0 ± 2.8	4.5 ± 2.6	0.042	0.914
Average snack	33.1 ± 18.1	36.0 ± 12.9	25.1 ± 13.7	35.9 ± 15.7	<0.001	0.014

^1^ Values are means ± SDs. Significant differences determined from General Lineal Models adjusted by energy intake.

## Data Availability

The data presented in this study are available on request from the corresponding author.

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
