# Peer review of "Dietary Intake and Sources of Added Sugars in Various Food Environments in Costa Rican Adolescents"

_nutrients, 2022, doi:10.3390/nu14050959_

Round 1
Reviewer 1 Report
It is very important to know which is the date of the database of the School of Human Nutrition of the University of Costa Rica. If it is an old database previously to the reformulation of foodstuff from food and beverage industry in Costa Rica it is to need to include this date and reference in the title of the paper
The table date contain data for food groups rich in added sugar g/d and the amounts are higher s total sugar intake and added sugar intake. Please, I need a clarification
Change vegetable-milk beverages by plant beverages
Change milk based beverages by milk and milk based beverages
Author Response
Dear Reviewer, thank you for your comments. Our responses are immediately below each statement.
1-It is very important to know which is the date of the database of the School of Human Nutrition of the University of Costa Rica. If it is an old database previously to the reformulation of foodstuff from food and beverage industry in Costa Rica it is to need to include this date and reference in the title of the paper.
Response 1. A relevant paragraph was added to section 2.4, Added Sugar Assessment. It includes details regarding the database of the School of Human Nutrition of the University of Costa Rica and its updates. The paragraph now reads:
Dietary intake of added sugars was estimated using the following resources: 1) recipes provided by the adolescents on their food records, as well as from a collection of recipes from dietary studies conducted in communities, available at the School of Human Nutrition of the University of Costa Rica; 2) the database of the School of Human Nutrition of the University of Costa Rica, which contains 1655 items from the following sources: 1307 foods from USDA, 80 foods from the food composition tables of the Institute of Nutrition for Central America and Panama (INCAP), 254 foods from recipes commonly consumed in Costa Rica (e.g. frescos, pastries, breads, cookies and desserts), 13 foods from the Costa Rican mandatory fortification food group, and updated information on foodstuffs from the food and beverage industry in Costa Rica as of 2017 (year in which the study data were collected) (46), including all the foods that were reformulated by the Costa Rican food industry to reduce added sugar content; 3) manufacturer's information, and 4) lists of ingredients and other labeling information.
2-The table date contain data for food groups rich in added sugar g/d and the amounts are higher s total sugar intake and added sugar intake. Please, I need a clarification.
Response 2. The information on Table 1 regarding food groups that are rich in added sugar refers to intake of various food groups (in g/person/day) and not to intake of sugars from such food groups. This was clarified on Table 1 and its captions.
3-Change vegetable-milk beverages by plant beverages.
Response 3. The term was amended in Figures 2 and 3.
4-Change milk based beverages by milk and milk based beverages
Response 4. The term was amended in Table 1 and Figures 2 and 3.

Reviewer 2 Report
The authors estimated added sugars intake and the food sources in Costa Rican adolescents. They also investigated whether the intake and good sources differed by social environments. Although the topic is potentially important and interesting, there are many serious concerns that should be sufficiently addressed to improve the quality of the manuscript.
1) The most serious concern of this study is the inconsistency of the main aim of this study. For example, the authors stated the main aims were to identify dietary intake and food sources of added sugars by social environments and determine the difference of added sugar consumption according to mealtimes or weekdays/weekends (lines 67-69). However, results according to mealtimes were not shown either in Figures 1 and 2 and mentioned in the discussion section. Additionally, although the discussion section should start with a summary of the main results, the first paragraph includes the results according to added sugars intake by region (urban vs rural) and SES, which are not the main aims. The authors should re-consider what they would like to investigate in this study.
2) Descriptions of estimating added sugars intake is also inaccurate or insufficient. While the authors explained they referred to recipes derived from food records, a database of the School of Human Nutrition of the University of Costa Rica, manufacturer's information, and information on food labels (lines 133-136), they also referred to the food composition database in lines 163-168. If the authors referred to the former information, what was the latter database used for? Further, for estimation of added sugars contents, recipes or information on the composition of ingredients are necessary for all food items, but collecting such information is impossible in many cases. Therefore, the authors may make some assumptions, or added sugars intake may not be estimated for some food items, although they did not provide any explanations for these issues.
3) Another serious concern is inappropriate statistical analysis, i.e. misuse of ANOVA because ANOVA cannot be used to test proportions and breakdowns. Therefore, the chai-square is appropriate for the results in Table 4 and Figures 1 and 2, except for those by social environments (and mealtimes), for which any statistical analysis may not be necessary. Support by a nutritional epidemiologist and a biostatistician may improve the quality of this study.
4) There are many mistakes in the results section. According to the descriptions in the text, Table 2 (line 204), Table 3 (line 212), and Table 4 (line 239) are considered to mention the results in Table 3, Table 4, and Table 2 respectively. The descriptions in lines 219-220 and 227-235 are not shown in any Tables and Figures. Figure 1 presents the same results in Figure 2. The authors should carefully check their manuscript before submission. Otherwise, readers may consider that the authors do not understand what they are doing.
Author Response
REVIEWER 2
Dear Reviewer, thank you for your comments. Our responses are immediately below each statement.
The authors estimated added sugars intake and the food sources in Costa Rican adolescents. They also investigated whether the intake and good sources differed by social environments. Although the topic is potentially important and interesting, there are many serious concerns that should be sufficiently addressed to improve the quality of the manuscript.
1) The most serious concern of this study is the inconsistency of the main aim of this study. For example, the authors stated the main aims were to identify dietary intake and food sources of added sugars by social environments and determine the difference of added sugar consumption according to mealtimes or weekdays/weekends (lines 67-69). However, results according to mealtimes were not shown either in Figures 1 and 2 and mentioned in the discussion section. Additionally, although the discussion section should start with a summary of the main results, the first paragraph includes the results according to added sugars intake by region (urban vs rural) and SES, which are not the main aims. The authors should re-consider what they would like to investigate in this study.
Response 1a. Results regarding added sugar consumption according to mealtime are presented on Table 4.
Response 1.b. Taking into consideration the comments of the reviewer regarding inconsistency in focus, the manuscript has been revised to make the aims more clear and consistently stated throughout the writing and the results. The discussion includes several paragraphs about results related to added sugar consumption at various mealtimes:
Pag. 20. It is of particular concern that, overall, 43% of adolescents consume over 20% TEI from added sugars and that much of this sugar is consumed primarily during breakfast, lunch, and snack mealtimes, and during weekend days.
Pag. 22. Changing the availability of frescos to promote a healthier diet in adolescents is a complex task because the ingredients to prepare them vary, it is difficult to regulate them in various environments, and their consumption at mealtimes is firmly embedded in the Costa Rican food culture, given the practice of drinking frescos with snacks, lunch and dinner or as thirst-quenchers [60].
Response 1c. A paragraph summarizing the main results of the study was added to the introduction. It now reads:
This study provides the first picture of added sugar intakes and sources in Costa Rican adolescents. The consumption of added sugar exceeds the recommendation established by the WHO in a large proportion (90%) of adolescents. The highest consumption occurs in the family environment, during breakfast, lunch, and snack mealtimes, and during weekend days.
Response 1d. To be consistent with the study aims, we removed content related to added sugar intake by region (urban vs rural) from the discussion. It now reads:
Like other studies, added sugar intake among Costa Rican adolescents is notably higher than the recommended < 10% TEI [15, 18-21], both on weekdays and weekends. It is of particular concern that, overall, 43% of adolescents consume over 20% TEI from added sugars and that much of this sugar is consumed primarily during breakfast, lunch, and snack mealtimes, and during weekend days.
2) Descriptions of estimating added sugars intake is also inaccurate or insufficient. While the authors explained they referred to recipes derived from food records, a database of the School of Human Nutrition of the University of Costa Rica, manufacturer's information, and information on food labels (lines 133-136), they also referred to the food composition database in lines 163-168. If the authors referred to the former information, what was the latter database used for? Further, for estimation of added sugars contents, recipes or information on the composition of ingredients are necessary for all food items, but collecting such information is impossible in many cases. Therefore, the authors may make some assumptions, or added sugars intake may not be estimated for some food items, although they did not provide any explanations for these issues.
Response 2a. A relevant paragraph was added to section 2.4, Added Sugar Assessment. It includes details regarding the database of the School of Human Nutrition of the University of Costa Rica and its updates. The paragraph now reads:
Dietary intake of added sugars was estimated using the following resources: 1) recipes provided by the adolescents on their food records, as well as from a collection of recipes from dietary studies conducted in communities, available at the School of Human Nutrition of the University of Costa
Rica; 2) the database of the School of Human Nutrition of the University of Costa Rica, which contains 1655 items from the following sources: 1307 foods from USDA, 80 foods from the food composition tables of the Institute of Nutrition for Central America and Panama (INCAP), 254 foods from recipes commonly consumed in Costa Rica (e.g. frescos, pastries, breads, cookies and desserts), 13 foods from the Costa Rican mandatory fortification food group, and updated information on foodstuffs from the food and beverage industry in Costa Rica as of 2017 (year in which the study data were collected) (46), including all the foods that were reformulated by the Costa Rican food industry to reduce added sugar content; 3) manufacturer's information, and 4) lists of ingredients and other labeling information. Although all these tools were used to estimate consumption of added sugars, it was impossible to collect precise information for every single case; therefore, estimates were made for some food items using values from similar foods, as has been suggested by Wanselius et al. [50].
Response 2b. Section 2.7 Data Analysis was rewritten to clarify the use of the food composition database. It now reads:
Epi Info™ software, version 3.5.4 (2008), was used to process data from the 3-day food records. The previously described database of the School of Human Nutrition of the University of Costa Rica was used with this tool.
3) Another serious concern is inappropriate statistical analysis, i.e. misuse of ANOVA because ANOVA cannot be used to test proportions and breakdowns. Therefore, the chi-square is appropriate for the results in Table 4 and Figures 1 and 2, except for those by social environments (and mealtimes), for which any statistical analysis may not be necessary. Support by a nutritional epidemiologist and a biostatistician may improve the quality of this study.
Response 3. Thank you for your feedback. We ran the data again using the chi-square test. Several p-values changed. We replaced the corresponding values on the text, on Table 3 (formerly Table 4) and on Figures 2 and 3.
4) There are many mistakes in the results section. According to the descriptions in the text, Table 2 (line 204), Table 3 (line 212), and Table 4 (line 239) are considered to mention the results in Table 3, Table 4, and Table 2 respectively. The descriptions in lines 219-220 and 227-235 are not shown in any Tables and Figures. Figure 1 presents the same results in Figure 2. The authors should carefully check their manuscript before submission. Otherwise, readers may consider that the authors do not understand what they are doing.
Response 4a. We sincerely apologize for the oversight. The tables have been correctly relocated and are consistent with the text.
Response 4b. The information under “Dietary intake of added sugars by social environment, mealtime and day of the week” it now presented in a new Figure 1.

Round 2
Reviewer 1 Report
It is right for me